# The Immunoscore in Localized Urothelial Carcinoma Treated with Neoadjuvant Chemotherapy: Clinical Significance for Pathologic Responses and Overall Survival

**DOI:** 10.3390/cancers13030494

**Published:** 2021-01-28

**Authors:** Elise F. Nassif, Bernhard Mlecnik, Constance Thibault, Marie Auvray, Daniela Bruni, Alexandre Colau, Eva Compérat, Gabriela Bindea, Aurélie Catteau, Aurélie Fugon, Isabelle Boquet, Marine Martel, Philippe Camparo, Pierre Colin, Roubini Zakopoulou, Aristotelis Bamias, Mostefa Bennamoun, Xavier Barthere, Bruno D’acremont, Marine Lefevre, Francois Audenet, Arnaud Mejean, Virginie Verkarre, Stéphane Oudard, Jérôme Galon

**Affiliations:** 1Oncology Department, Hopital Européen Georges Pompidou, University of Paris, 75015 Paris, France; elise.nassif@lyon.unicancer.fr (E.F.N.); constance.thibault@aphp.fr (C.T.); marie.auvray2@aphp.fr (M.A.); 2Centre de Recherche des Cordeliers, Laboratory of Integrative Cancer Immunology, INSERM, Université de Paris, 75015 Paris, France; bernhard.mlecnik@crc.jussieu.fr (B.M.); danielabruni039@gmail.com (D.B.); gabriela.bindea@crc.jussieu.fr (G.B.); jerome.galon@crc.jussieu.fr (J.G.); 3Inovarion, 75005 Paris, France; 4Urology Department, Diaconnesses Hospital, 75020 Paris, France; AColau@hopital-dcss.org; 5Pathology Department, Diaconnesses Hospital, 75012 Paris, France; eva.comperat@aphp.fr; 6HalioDx, 13009 Marseille, France; aurelie.catteau@haliodx.com (A.C.); aurelie.fugon@haliodx.com (A.F.); isabelle.boquet@haliodx.com (I.B.); marine.martel@haliodx.com (M.M.); 7Centre de Pathologie, 80000 Amiens, France; p.camparo@anapath.fr; 8Urology Department, La Louvière, 59042 Lille, France; pierre_colin@msn.com; 9Oncology Department, Alexandra Hospital, 11528 Athens, Greece; rzakopoul@gmail.com (R.Z.); abamias@med.uoa.gr (A.B.); 10Oncology Department, Institut Mutualiste Montsouris, 75014 Paris, France; mostefa.bennamoun@imm.fr (M.B.); xavier.barthere@gustaveroussy.fr (X.B.); 11Urology Department, Clinique St Jean De Dieu, 75007 Paris, France; brdacrem@club-internet.fr; 12Department of Pathology, Institut Mutualiste Montsouris, 75014 Paris, France; marine.lefevre@imm.fr; 13Urology Department, Hopital Européen Georges Pompidou, AP-HP—Université de Paris, 75015 Paris, France; francois.audenet@gmail.com (F.A.); virginie.verkarre@aphp.fr (V.V.); 14Pathology Department, Hopital Européen Georges Pompidou, 75015 Paris, France; arnaud.mejean@aphp.fr

**Keywords:** urothelial carcinoma, neoadjuvant chemotherapy, Immunoscore, prognostic score, predictive response to chemotherapy, immuno-oncology

## Abstract

**Simple Summary:**

The prognosis of localized muscle-invasive bladder cancer is poor and the prognostic and predictive biomarkers of the response to treatment are lacking. We retrospectively investigated the role of the Immunoscore in the neoadjuvant setting, where the Immunoscore is a standardized quantitative assay of lymphocytes in the tumor microenvironment. We found it allowed for the stratification of patient prognoses and the prediction of response to neoadjuvant chemotherapy.

**Abstract:**

(1) Background—The five-year overall survival (OS) of muscle-invasive bladder cancer (MIBC) with neoadjuvant chemotherapy and cystectomy is around 50%. There is no validated biomarker to guide the treatment decision. We investigated whether the Immunoscore (IS) could predict the pathologic response to neoadjuvant chemotherapy and survival outcomes. (2) Methods—This retrospective study evaluated the IS in 117 patients treated using neoadjuvant chemotherapy for localized MIBC from six centers (France and Greece). Pre-treatment tumor samples were immunostained for CD3+ and CD8+ T cells and quantified to determine the IS. The results were associated with the response to neoadjuvant chemotherapy, time to recurrence (TTR), and OS. (3) Results—Low (IS-0), intermediate (IS-1–2), and high (IS-3–4) ISs were observed in 36.5, 43.7, and 19.8% of the cohort, respectively. IS was positively associated with a pathologic complete response (pCR; *p*-value = 0.0096). A high IS was found in 35.7% of patients with a pCR, whereas it was found in 11.3% of patients without a pCR. A low IS was observed in 48.4% of patients with no pCR and in 21.4% of patients with a pCR. Low-, intermediate-, and high-IS patients had five-year recurrence-free rates of 37.2%, 36.5%, and 72.6%, respectively. In the multivariable analysis, a high IS was associated with a prolonged TTR (high vs. low: *p* = 0.0134) and OS (high vs. low: *p* = 0.011). (4) Conclusions—This study showed the significant prognostic and predictive roles of IS regarding localized MIBC.

## 1. Introduction

Bladder cancer is the fourth and eighth most common cancer in men and women respectively, in Western countries [1]. Approximately 30% of patients will present with muscle-invasive bladder cancer (MIBC). The standard-of-care treatment for patients with localized MIBC is neoadjuvant chemotherapy before a radical cystectomy [2,3]. The five-year overall survival (OS) rate after this initial treatment is around 50% [4]. Major clinical prognostic factors are the tumor T stage and nodal status [5]. The pathologic complete response (pCR) to neoadjuvant chemotherapy is associated with an improved OS [6]. There is no tumor biomarker that is validated for the prediction of the response to chemotherapy or prognosis in this context.

Urothelial carcinoma (UC) is the most frequent histologic type of bladder cancer and is known for its immunogenicity. The importance of stimulating immune responses against the tumor has long been exploited by the use of BCG (Bacille Calmette et Guérin) instillations in the treatment of non-muscle-invasive UC. The lymphocytic infiltrate seems to play a major role in BCG therapy’s efficacy [7]. In the metastatic setting, stimulation of the immune adaptive system with checkpoint inhibitors, namely, programmed death receptor-1 (PD-1) and programmed death ligand-1 (PD-L1) inhibitors, such as atezolizumab [8], avelumab [9], durvalumab [10], nivolumab [11], and pembrolizumab [12], leads to objective response rates (ORRs) of 20–25%. These drugs are now recommended in the treatment of metastatic UC, either in the second line after chemotherapy or in the first line for cisplatin unfit patients with a PD-L1+ tumor or for patients that are ineligible for platinum-based chemotherapy, regardless of tumor PD-L1 status. Avelumab is recommended as maintenance therapy in patients that have not progressed with first-line platinum-containing chemotherapy [13].

Immune cells that are present in the microenvironment play a major role in slowing down tumor progression [14]. A strong lymphocytic infiltration has been associated with longer overall survival and progression-free survival in various cancer types [15]. The Immunoscore (IS) is a consensus immune tumor prognostic biomarker that is currently under investigation for multiple tumor types [16,17,18]. This standardized immune score assesses the density of lymphocytic T infiltrates (CD3/CD8), as well as its site (in the invasive margins or in the center of the tumor). This score has demonstrated its superiority over the TNM (Tumor Node Metastasis) classification in colorectal cancer [16,19] and locally advanced colorectal cancer [20,21,22], and is referenced in the WHO classification and recommended by ESMO (European Society of Medical Oncology) clinical practice guidelines for localized colorectal cancer [23]. It is also available as an in vitro diagnostic test for clinical use (CE-IVD) in colon cancer. 

We sought to investigate the prognostic and predictive role of the Immunoscore in localized MIBC patients undergoing neoadjuvant chemotherapy.

## 2. Materials and Methods

Patients treated in five centers in France (HEGP: Hôpital Européen Georges Pompidou, Hôpital des Diaconnesses, Clinique La Louvière, Clinique St Jean De Dieu, Institut Mutualiste Montsouris) and one center in Greece (Alexandra Hospital) for localized MIBC between January 2003 and December 2016 were included in this analysis. The inclusion criteria were as follows: histologically confirmed muscle-invasive UC (clinical stages II–IIIA), neoadjuvant platinum-based chemotherapy, and pre-chemotherapy sample available. The exclusion criteria were as follows: a metastatic disease at diagnosis, any histology other than UC, patient’s refusal to participate, and absence of survival data due to lack of a follow-up just after local treatment.

Clinical data were retrospectively collected and anonymized at each participating center prior to the central collection for analysis. Collected clinical data were related to the patients’ characteristics (sex, age at diagnosis, and risk factors for UC, such as tobacco use, professional exposure, and history of cancer with chemotherapy or pelvic irradiation) and the initial disease (primary site of the bladder and/or upper tract, TNM stage, and histologic variant). Data regarding the treatment modalities, response to treatment, and survival were also recorded for the outcome analysis. Tumor staging (clinical and pathological) was reported according to the AJCC (American Joint Comitte on Cancer) [24]. Clinical staging was done using computed tomography scans for the T and N stages. For the T stage, the pathological report of TURB (Trans-urethral resection of bladder) was used to indicate that the patient was at least in stage T2 and further assessment was done using radiological assessment. Recurrence was defined as either metastatic disease on the radiological assessment if seen within the first year of follow-up or histologically confirmed if recurrence was seen after 12 months following surgery, or local relapse, which was defined as histologically confirmed muscle-invasive urothelial carcinoma. The treatment of advanced disease was not recorded. The pCR was evaluated by specialized uropathologists at each participating center.

The initial tumor samples were either trans-urethral resections of bladder tumors containing muscle infiltration or biopsies of upper-tract UC lesions. Freshly cut slides of 4 μm on Superfrost Plus coating were used for the evaluation of each initial tumor sample. The slides had to be cut from archived pathology FFPE (Formalin-Fixed Paraffin-Embedded) blocks within six months before analysis. Immunostaining for CD3+ and CD8+ T lymphocytes were done at the INSERM UMRS (Institut National de la Santé et de la Recherche Médicale, Unité Mixte de Rechreche) 1138 laboratory, Paris, France.

Digital pathology was used to quantify densities of CD3+ and CD8+ T-lymphocytes in the tumor center (CT) and invasive margins (IM). The densities were calculated in terms of the number of positive cells per square millimeter. The CD3 and CD8 densities in CT and IM regions were converted into percentiles and the mean of the four percentiles obtained were calculated and translated into the Immunoscore scoring system. Groups were then determined according to the densities of the two lymphocyte populations (CD3 and CD8) in the two localizations (CT and IM), starting from IS-0 (low density of both populations in both localization) to IS-4 (elevated densities of both populations in both localization). The density thresholds were determined using the optimal cut-off method. An Immunoscore adapted to biopsies (ISb) was performed since no tumor invasive margins were identified in 21 specimens; three groups based on the densities of the two populations in the core tumor were determined: ISb-0 (low density of both populations), ISb-1 (low density of one population), and ISb-2 (high density of both populations). Clinical and treatment data were blinded to the investigators taking part in the Immunoscore staining and quantification.

When analyzing two groups high versus low, the expected proportion of patients in each was assumed to be at 50%. Applying Schoenfeld’s procedure in the case of MIBC with 60% of observed relapse events with a power of 90% and an alpha level at 5% (two-tail) and a hazard ratio (HR) of 1.85–2.1, the required sample size was estimated to be between 103 and 152 patients. The final sample size to determine a significant difference for the Immunoscore in MIBC was 117 patients after the clinical and biomarker data exclusion criteria were applied (Appendix A).

Fisher’s exact tests were applied to determine the associations between clinical characteristics, treatment procedures, and the Immunoscore. Student’s *t*-tests were used to evaluate the difference between the immune densities of each lymphocyte population within the clinical categories. All statistical tests were two-sided. Survival analysis was performed using log-rank tests and Cox proportional hazards models (survival, R package). The bivariable association of the Immunoscore stratified by participating center and the time-to-event outcomes were evaluated using Cox proportional hazards models. An alternative measure of the survival time distribution that was independent of the proportional hazards assumption was applied using the restricted mean survival time (RMST) as two-sample comparisons (survRM2, R package). The relative importance of each parameter to the survival risk was assessed using the chi-squared proportion (*χ*^2^) (rms, R package).

This study was conducted in accordance with the principles laid by the 18th World Medical Assembly (Helsinki, 1964). Written consent for the use of the patients’ data for analysis was encouraged but was not mandatory. Patients who could not provide written consent received the relevant information by post in order to enable denial on their behalf. Since data were anonymized, a waiver of consent was granted in the case of death, but families were informed and could deny participation. The data collection was declared at the French national commission (CNIL, Commission Nationale de l’informatique et des libertés, declaration 5G52360236q). Permission was granted by a French ethics review board (CEREES, Comité d’éthique de la recherche en santé, dossier no. 27756).

## 3. Results

### 3.1. Patients and Tumor Characteristics

A total of 117 patients treated with neoadjuvant platinum-based chemotherapy for invasive urothelial carcinoma were included in the final analysis (Appendix A). The mean age at diagnosis was 66.4 years old (SD: ±8.2) and 22.2% (*n* = 26) of the patients were women (Table 1). The major clinical and tumor characteristics at diagnosis are reported in Table 1. The median follow-up times (95% CI) were 26.7 months (24.1–36.8), and 31.4 months (24.1–46.7) for TTR and OS, respectively. The most used regimen was MVAC (Methotrexate, Vinblastine, Adriamycin, and Cisplatin). The second most prescribed regimen was GC (Gemcitabine and Cisplatin). The median number of cycles was four cycles (IQR = 4–5). 

A pCR was found in 35 patients (29.9%) after the neoadjuvant chemotherapy, whereas 66 patients (56.4%) had a persistent viable tumor and this information was lacking in pathology reports for 16 patients (13.7%).

### 3.2. Immune Densities and Immunoscore Distributions in Terms of the Clinical Characteristics

Thirty-eight patients (32.5%) were classified into the IS-0 group, 21 patients (17.9%) into IS-1, 25 patients (21.4%) into IS-2, 11 patients (9.4%) into IS-3, 8 patients (6.8%) into I-4, and 14 patients were not classified due to a lack of IM. Therefore, an optimal ISb using only core tumor densities was created, which involved three groups: ISb-0 with 43 patients (36.8%), ISb-1 with 26 patients (22.2%), and ISb-2 with 47 patients (40.2%). There was no association between IS or ISb and the patient characteristics. No significant association was found regarding the T or N stages, neither with the Immunoscore nor with the immune densities.

Notably, tumors with variant histologies (*n* = 17/117) were less infiltrated by lymphocytes in the IM for both lymphocyte populations (*t*-test, *p*-value < 0.05). There were only typical UC histology tumors in the high IS groups (IS3–4) (Figure 1A,B).

### 3.3. Comparative Outcomes of Immunoscore Groups in Terms of the Response to Treatment

The Immunoscore was significantly associated with obtaining a pCR after neoadjuvant chemotherapy. IS-3–4 was found in 35.7% of patients with a pCR, whereas it was found in only 11.3% of patients without a pCR. In contrast, an IS-0 Immunoscore was observed in 48.4% of patients with no pCR, and in only 21.4% of patients with a pCR (Fisher’s exact test, *p*-value = 0.0096; Table 2). Likewise, ISb efficiently stratified patients for response to neoadjuvant chemotherapy (Fisher’s exact test, *p*-value = 0.0035; Table 2). Patients without a pCR had an ISb-0 in 51.5% (*n* = 34) of cases and ISb-2 in 31.8% (*n* = 21) of cases. In contrast, patients with a pCR had an ISb-0 in 17.6% (*n* = 6) of cases and ISb-2 in 55.9% (*n* = 19) of cases.

### 3.4. Recurrence and Time to Recurrence

During follow-up, 65 patients (55.6%) exhibited recurrence, while this data was missing for 7 patients (6%). The median TTR of the whole cohort was 58.3 months (95% CI = 20.9–not reached (NR)). The N stage was significantly associated with the TTR (stratified Wald *p*-value = 0.0363; Table 3, Appendix A): the median TTR was 11.4 months versus NR with and without nodal involvement (HR = 4.21; 95% CI = 1.1–16.16). Patients without a pCR had a significantly shorter median TTR of 14.5 months versus NR. The five-year recurrence-free rate was of 23.2% vs. 82.9% in the no-pCR and pCR groups, respectively (HR = 0.12, 95% CI = 0.04–0.34, stratified Wald *p*-value ≤ 0.0001).

Regarding the immune infiltration, the CD8 density in the tumor center was associated with an improved TTR (stratified Wald *p*-value = 0.0063): the high CD8-ct infiltration group had a median TTR of NR versus 17.7 months in the CD8-ct low group (HR = 0.41; 95% CI = 0.22–0.78). The IS-0 group had a significantly shorter median TTR of 16.4 months versus NR in the IS-3–4 group. The five-year recurrence-free rates were 37.2, 36.5, and 72.6% in the IS-0, IS-1–2, and IS-3–4 groups, respectively. Comparing the IS-3–4 group and the IS-0 group, the HR was 0.14 (95% CI = 0.03–0.64, stratified Wald *p*-value = 0.011; Table 3, Figure 2, Appendix A).

The optimized biopsy ISb scoring system allowed for stratifying patients efficiently for the TTR: 34.2% of patients were recurrence-free at five years in the ISb-0 group compared to 53.9% in the ISb-1–2 group (HR = 0.42, 95% CI = 0.23–0.79, stratified Wald *p*-value = 0.007; Table 3, Figure 2, Appendix A). ISb-0 and ISb-1–2 had median TTRs of 14.5 months and 61.6 months.

### 3.5. Overall Survival

The median OS of the whole cohort was 55.3 months (95% CI = 32.9–74.8); this information was missing for eight patients. Notably, patients with nodal involvement had a median OS of 25.8 months compared to 36.8 months for the N0 patients (stratified Wald *p*-value = 0.47; Appendix A). Patients with no pCR had a significantly shorter median OS of 46.5 months versus NR in the pCR group (HR = 0.28, 95% CI = 0.11–0.68, stratified Wald *p*-value = 0.005).

CD8-ct infiltration was significantly associated with OS (stratified Wald *p*-value = 0.002): the high CD8-ct infiltration group had a median OS of 64.7 months versus 27.4 months in the low CD8-ct group. Likewise, the high CD3-ct group had a median OS of 64.7 months versus 36.8 months in the low CD3-ct group (stratified Wald *p*-value = 0.013).

The IS-0 group had a significantly shorter median OS of 27.4 months versus NR in the IS-3–4 group. Comparing the IS-3–4 group and the IS-0 group, the HR was 0.22 (95% CI = 0.06–0.78, *p*-value = 0.02; Appendix A, Figure 2 and Appendix A). The modified biopsy scoring system was also very efficient at stratifying patients for OS with ISb-0 and ISb-1–2 having median OSs of 27.4 months and 60.5 months, respectively. Moreover, the ISb-0 and ISb-1–2 group had a five-year OS rate of 39.7% and 53.2%, respectively (HR = 0.3, 95% CI = 0.15–0.59, stratified Wald *p*-value = 0.0005; Appendix A, Figure 2 and Appendix A). Other subgroup analysis results displayed a non-significant beneficial effect of a higher Immunoscore on OS.

### 3.6. Multivariable Analysis

In the multivariable analysis stratified by center, IS-3–4 groups were significantly and independently associated with an improved TTR with an HR of 0.14 (95% CI = 0.03–0.67, Wald *p*-value = 0.013; Appendix A). Likewise, the IS groups were significantly and independently associated with an improved OS. The IS-3–4 groups had a HR for OS of 0.17 compared to IS-0 (95% CI = 0.04–0.66, Wald *p*-value = 0.011; Appendix A). The Immunoscore showed the strongest contribution of *χ*^2^ proportion among all other variables for influencing survival (TTR and OS; Figure 2 and Appendix A).

The modified ISb scoring system was significantly and independently associated with both TTR and OS. ISb-2 had an HR of 0.38 (95% CI = 0.18–0.81, Wald *p*-value = 0.012; Table 4) for TTR compared to ISb-0, and an HR of 0.21 (95% CI = 0.08–0.51, Wald *p*-value = 0.0006; Table 4) for OS compared to ISb-0.

## 4. Discussion

This study investigated the potential prognostic and predictive role of the Immunoscore in 117 patients with localized MIBC undergoing neoadjuvant chemotherapy before local treatment. We found that the Immunoscore was associated with the pCR, TTR, and OS.

The Immunoscore is a validated strong prognostic factor in colorectal cancer [25] and a predictor for the response to chemotherapy in this setting [20,21]. The immune contexture of tumors and the Immunoscore have been demonstrated to be strong predictors in malignant tumors and have been proposed as new tools to classify tumors [15,26,27,28,29]. In localized colorectal cancer, the Immunoscore was found to be a more precise prognostic tool than TNM staging [26] and microsatellite instability [25].

Although our study is retrospective and is limited by missing clinical data, such as N-stage data, this is a real-life multicentric large cohort report on standard-of-care treatment with neoadjuvant platinum-based chemotherapy for MIBC. In concert with previous reports, we found that clinical N+ and higher T stages were significantly associated with a worse TTR [5,30], while the pCR rate reported in the literature being roughly 30% and correlated with the OS and TTR is also consistent with our data [6]. A higher clinical stage was not statistically significant in the multivariable analysis, which is probably due to the difficulty in achieving correct staging using radiological evaluation. Although this represents routine standard-of-care daily practice, it likely underestimates some patients’ staging. 

The importance of immune infiltration, particularly CD8+ T lymphocytes, has been consistently found to be a prognostic tool for advanced UC [31]. In non-muscle-invasive UC, CD3 and CD8 lymphocytic infiltrates, stratified by localization in the tumor, have been recently associated with prognosis [32]. The prognostic impact of immune parameters in localized MIBC has been reported in other studies, with a positive association between lymphocyte infiltration and survival [33,34]. Immunoscore, CD3-im, CD3-ct, CD8-im, and CD8-ct lymphocytes have been previously evaluated in MIBC on cystectomy blocks [35]. The authors reported on the prognostic impact of CD3 and CD8 lymphocytes at the IM, whereas we found a more pronounced prognostic role of lymphocytes in the core tumor. This difference could be explained by the neoadjuvant treatment effect on the immune microenvironment and difficulties encountered when studying IM in a transurethral resection of the bladder samples. Differences in the patient selection (we only included MIBC patients treated with neoadjuvant chemotherapy), as well as methodological differences (we performed the standardized Immunoscore assay) may account for these discrepancies.

We report a difference in immune densities and the Immunoscore between classical UC and variant histologies, which has already been suggested [36,37]. The tumor microenvironment, the Immunoscore, and immune contexture vary according to UC subtypes and variants, and their prognostic and predictive value in these specific settings warrants further assessment [38]. Interestingly, the CD3 and CD8 infiltration has also been associated with the tumor histologic grade [32]. The aforementioned reports point toward heterogeneity in the distribution of CD3 and CD8 lymphocyte densities across subtypes and settings. However, each separate study shows a specific predominant prognostic impact of one of the individual densities of either CD3-ct, CD3-im, CD8-ct, or CD8-im. As these variants are rare, specific studies of these histological subtypes should be proposed in order to properly assess their impact. A combined consensus reproducible assay, such as the Immunoscore, could be more robust when performing across subtypes and clinical stages.

As new trials are testing immunotherapy with or without chemotherapy in the same setting [39,40,41], the role of the Immunoscore to efficiently predict responses to standard-of-care treatment seems interesting. The ultimate goal will be to stratify patients who will benefit most from the combination therapy, or from a monotherapy of either checkpoint inhibitors or chemotherapy. Assessing the benefits and risks of each modality treatment in this curative setting is essential. Notably, some trials in the advanced setting were negative, in part due to a statistical analysis based on PDL1 expression only [42]. This underscores the importance of finding efficient predictive immune biomarkers in this disease.

## 5. Conclusions

The Immunoscore efficiently allows for patient risk stratification and prediction of responses to neoadjuvant chemotherapy. These results warrant further evaluation, which is underway in prospective trials before being implemented in routine clinical practice.

## Figures and Tables

**Figure 1 cancers-13-00494-f001:**
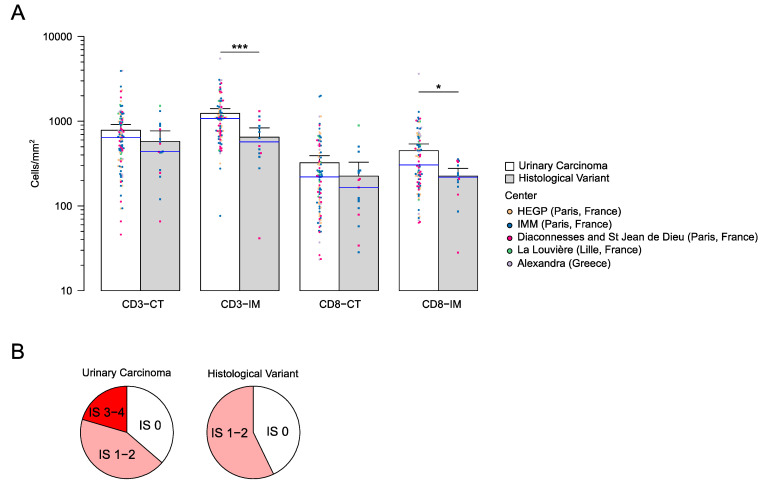
The immune infiltrate and histological variants: (**A**) patient groups comparing the immune densities in classical urothelial carcinoma to other variants and (**B**) distribution of the Immunoscore in urothelial carcinoma and other variants. * *p*-value < 0.05, *** *p*-value < 0.005.

**Figure 2 cancers-13-00494-f002:**
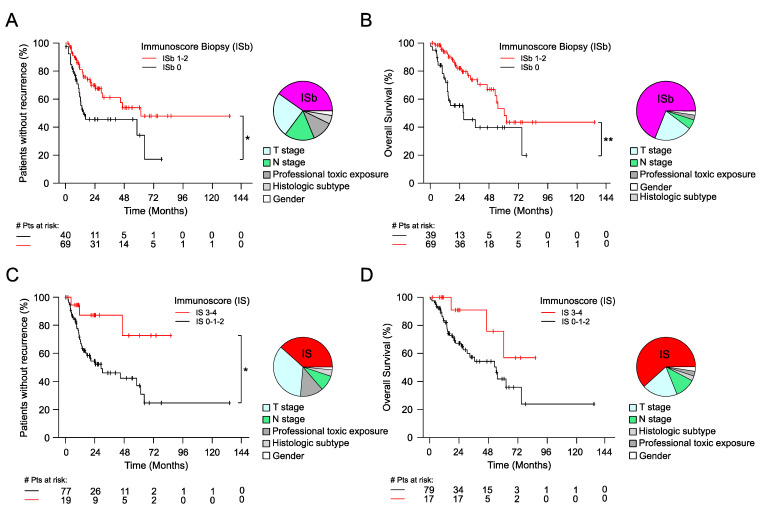
The impact of Immunoscore on patient outcome. Kaplan–Meier curves for the Immunoscores are shown for time to recurrence (TTR) (**A**,**C**) and overall survival (OS) (**B**,**D**). The pie chart indicates the contribution of *χ*^2^ proportion of ISb and IS to other variables for influencing the survival in the multivariate analysis. (**A**,**B**) Two Immunoscore categories for the biopsy Immunoscores: ISb-0 (black) and ISb-1–2 (red). (**C**,**D**) Two Immunoscore categories for Immunoscore: IS-0–2 (black) and IS 3–4 (red). * *p* < 0.05, ** *p* < 0.01.

**Table 1 cancers-13-00494-t001:** Clinical characteristics.

Clinical Characteristics		Cohort (*N* = 117)
Age at surgery (years)		
	*N*	117 (100%)
	Mean (SD)	66.4 (8.2)
	Range	41–82
Center		
	France: La Louvière (Lille)	9 (7.7%)
	France: HEGP (Paris)	20 (17.1%)
	France: IMM (Paris)	49 (41.9%)
	France: Diaconnesses and St Jean de Dieu (Paris)	27 (23.1%)
	Greece: Alexandra	12 (10.3%)
Gender		
	Male	91 (77.8%)
	Female	26 (22.2%)
Professional toxic exposure		
	No	75 (64.1%)
	Yes	10 (8.5%)
	Not available	32 (27.4%)
Tobacco use		
	No	27 (23.1%)
	Yes	78 (66.7%)
	Not available	12 (10.3%)
Previous cancer with chemotherapy or pelvic radiotherapy		
	No	106 (90.6%)
	Yes	6 (5.1%)
	Not available	5 (4.3%)
Site of primary tumor		
	Bladder	115 (98.3%)
	Bladder + upper tract	1 (0.9%)
	Upper tract	1 (0.9%)
Histologic variant		
	Urothelial carcinoma	99 (84.6%)
	Variant	17 (14.5%)
	Not Available	1 (0.9%)
T stage		
	T2	106 (90.6%)
	T3–4	7 (6%)
	Not available	4 (3.4%)
N stage		
	N0	31 (26.5%)
	N+	77 (65.8%)
	Not Available	9 (7.7%)
Prior BCG therapy		
	No	93 (79.5%)
	Yes	24 (20.5%)
Creatinine clearance (MDRD)		
	*n*	44 (37.6%)
	Mean (SD)	81.8 (20.2)
	Range	49–150
	Not Available	73 (62.4%)
Neoadjuvant chemotherapy type		
	MVAC	66 (56.4%)
	GC	44 (37.6%)
	Carboplatin based	3 (2.5%)
	Other platinum-based	4 (3.4%)
pCR		
	No	66 (56.4%)
	Yes	35 (29.9%)
	Not Available	16 (13.7%)

HEGP: Hôpital Européen Georges Pompidou, IMM: Institut Mutualiste Montsouris, BCG: Bacille Calmette et Guérin, MDRD: Modification of Diet in Renal Diseases, MVAC: Methotrexate, Vinblastine, Adriamycine, and Cisplatin; GC: Gemcitabine and Cisplatin; pCR: pathologic complete response.

**Table 2 cancers-13-00494-t002:** Pathological complete response distribution according to the Immunoscore categories.

Immunoscore		No. of Patients without a pCR (%)	No. of Patients with a pCR (%)	*p*-Value *
Total		66	34	
ISb 3 groups				0.0035
	ISb-0	34 (51.5%)	6 (17.6%)	
	ISb-1	11 (16.7%)	9 (26.5%)	
	ISb-2	21 (31.8%)	19 (55.9%)	
ISb 2 groups				0.0012
	ISb-0	34 (51.5%)	6 (17.6%)	
	ISb-1–2	32 (48.5%)	28 (82.4%)	
Total #		62	28	
IS 3 groups				0.0096
	IS-0	30 (48.4%)	6 (21.4%)	
	IS-1–2	25 (40.3%)	12 (42.9%)	
	IS-3–4	7 (11.3%)	10 (35.7%)	
IS 2 groups				0.0093
	IS-0–2	55 (88.7%)	18 (64.3%)	
	IS-3–4	7 (11.3%)	10 (35.7%)	

* Fisher’s exact *p*-value, IS: Immunoscore, ISb: Immunoscore adapted to biopsies, # 14 patients were not classified due to missing invasive margins.

**Table 3 cancers-13-00494-t003:** Bivariable analysis for the clinical parameters for the time to recurrence.

Variable		Number of	Median Months	Rate at		Unadjusted Stratified by Center			Restricted Mean Survival Time	
		Patients (%)	(95% CI)	3 yr % (95% CI)	5 yr % (95% CI)	HR (95% CI)	*p*-Value *	C-Index (95% CI)	Rel. Months (95% CI)	*p*-Value **
Gender								0.52 (0.46–0.57)		
	Male	87 (79.1)	61.6 (20–NR)	58.4 (47.5–71.6)	51 (38.6–67.5)	1.0 (reference)			0.0 (reference)	
	Female	23 (20.9)	29.2 (9.4–64.5)	42.6 (23–78.8)	28.4 (10.3–77.9)	1.47 (0.73–2.95)	0.2845		−8.9 (−22.2 to 4.5)	0.1920
Previous cancer with chemotherapy or pelvic radiotherapy								0.5 (0.47–0.54)		
	No	99 (94.3)	58.3 (20–NR)	57.2 (47–69.5)	48 (36.3–63.6)	1.0 (reference)			0.0 (reference)	
	Yes	6 (5.7)	29.2 (3.4–NR)	33.3 (7.5–100)	NR (NR–NR)	1.29 (0.39–4.3)	0.6774		−5.3 (−23.8 to 13.1)	0.5697
Professional toxic exposure								0.54 (0.48–0.6)		
	No	74 (91.4)	44.9 (14.5–64.5)	51.9 (40.3–66.9)	48.7 (36.7–64.6)	1.0 (reference)			0.0 (reference)	
	Yes	7 (8.6)	NR	85.7 (63.3–100)	64.3 (33.8–100)	0.4 (0.09–1.7)	0.2128		22.1 (0.6–43.6)	0.0435
Tobacco use								0.49 (0.41–0.58)		
	No	27 (27.3)	29.2 (14.2–NR)	50 (33.1–75.6)	43.7 (26.8–71.4)	1.0 (reference)			0.0 (reference)	
	Yes	72 (72.7)	NR	57.6 (45.6–72.8)	50.4 (35.5–71.6)	0.93 (0.48–1.81)	0.8289		6.2 (−10.3 to 22.7)	0.4642
Histologic variant								0.5 (0.43–0.57)		
	UC	92 (84.4)	46.5 (20–NR)	54.6 (43.8–68.1)	48.9 (37.3–63.9)	1.0 (reference)			0.0 (reference)	
	Variant	17 (15.6)	58.3 (7.3–NR)	57.8 (37–90.3)	38.5 (15.4–96.3)	1.19 (0.54–2.66)	0.6656		0 (−15.5 to 15.4)	0.9964
T stage								0.51 (0.48–0.53)		
	T2	99 (90)	61.6 (16.4–NR)	59.3 (48.9–71.8)	53.6 (42.3–68)	1.0 (reference)			0.0 (reference)	
	T3–4	7 (6.4)	20.9 (6.3–NR)	NR (NR–NR)	NR (NR–NR)	1.93 (0.71–5.27)	0.1979		−0.4 (−5.7 to 4.8)	0.8664
N stage								0.6 (0.46–0.74)		
	N0	31 (79.5)	NR	69.3 (52.1–92.1)	69.3 (52.1–92.1)	1.0 (reference)			0.0 (reference)	
	N+	8 (20.5)	11.4 (2.3–29.2)	15 (2.5–90.6)	NR (NR-NR)	4.21 (1.1–16.16)	0.0363		−17.4 (−30.7 to −4.1)	0.0103
IS biopsy (2 groups)								0.63 (0.55–0.72)		
	ISb-0	40 (36.7)	14.5 (10.4–58.3)	45.6 (31.4–66.1)	34.2 (17.4–67.3)	1.0 (reference)			0.0 (reference)	
	ISb-1–2	69 (63.3)	61.6 (24.4–NR)	61.1 (48.6–76.8)	53.9 (40.5–71.9)	0.42 (0.23–0.79)	0.0072		14.6 (0.7–28.4)	0.0391
IS biopsy (3 groups)								0.63 (0.55–0.72)		
	ISb-0	40 (36.7)	14.5 (10.4–58.3)	45.6 (31.4–66.1)	34.2 (17.4–67.3)	1.0 (reference)			0.0 (reference)	
	ISb-1	24 (22)	NR	62.8 (43.2–91.3)	53.8 (33.3–87.1)	0.38 (0.16–0.91)	0.0299		16.3 (−1.6 to 34.2)	0.0751
	ISb-2	45 (41.3)	61.6 (20–NR)	59.6 (44.4–80.1)	53.7 (37.4–77)	0.44 (0.22–0.88)	0.0203		13.5 (−1.8 to 28.8)	0.0839
Immunoscore (2 groups)								0.6 (0.53–0.66)		
	IS-0–2	77 (80.2)	29.2 (13.7–58.3)	46.1 (34.3–62.1)	37 (24–56.9)	1.0 (reference)			0.0 (reference)	
	IS-3–4	19 (19.8)	NR	87.2 (71.9–100)	72.6 (48.4–100)	0.16 (0.04–0.69)	0.0135		29.7 (11.5–47.9)	0.0014
Immunoscore (3 groups)								0.65 (0.56–0.75)		
	IS-0	35 (36.5)	16.4 (10.4–64.5)	49.6 (34.3–71.9)	37.2 (18.9–73.2)	1.0 (reference)			0.0 (reference)	
	IS-1–2	42 (43.7)	29.2 (13.7–61.6)	42.6 (26.7–67.9)	36.5 (20.9–63.7)	0.81 (0.41–1.59)	0.5377		0.4 (−15.7 to 16.5)	0.9610
	IS-3–4	19 (19.8)	NR	87.2 (71.9–100)	72.6 (48.4–100)	0.14 (0.03–0.64)	0.0107		26.4 (8.4–44.3)	0.0040

* Wald *p* Value stratified by participating center. ** Restricted Mean Survival Time *p* value. IS: Immunoscore; ISb Immunoscore biopsy like.

**Table 4 cancers-13-00494-t004:** Multivariable analysis immunoscore biopsy vs. clinical parameters for time to recurrence and overall survival.

		TTR Model (45/109) *			OS Model (39/108) *		
	Variable	Hazard Ratio (95% CI)	*p*-Value ^1^	C-Index (95% CI)	Hazard Ratio (95% CI)	*p*-Value ^1^	C-Index (95% CI)
Multivariable Cox model stratified by center				0.7			0.73
				(0.61–0.8)			(0.64–0.82)
Immunoscore (3 groups)							
	ISb-1 vs. ISb-0	0.24 (0.08–0.7)	0.0087		0.25 (0.09–0.71)	0.0087	
	ISb-2 vs. ISb-0	0.38 (0.18–0.81)	0.0123		0.21 (0.08–0.51)	0.0006	
Gender							
	Female vs. male	1.37 (0.65–2.89)	0.4108		0.72 (0.29–1.77)	0.4716	
Professional toxic exposure							
	Yes vs. no	0.34 (0.07–1.63)	0.1754		0.79 (0.2–3.07)	0.7326	
	Unkown vs. no	0.69 (0.21–2.21)	0.5296		0.62 (0.17–2.32)	0.4815	
Histologic variant							
	Variant vs. UC	0.6 (0.22–1.66)	0.3278		1.12 (0.42–2.97)	0.8251	
	Unkown vs. UC	0 (0-Inf)	0.9977		-	-	
T stage							
	T3–4 vs. T2	2.8 (0.89–8.81)	0.0786		4.31 (1.04–17.76)	0.0434	
	Unknown vs. T2	3.08 (0.68–13.94)	0.1444		0.69 (0.12–4)	0.6750	
N stage							
	N+ vs. N0	4.14 (0.95–18.07)	0.0587		1.91 (0.31–11.76)	0.4839	
	Unknown vs. N0	1.48 (0.29–7.55)	0.6344		2.1 (0.37–12.06)	0.4044	

* Events/total; ^1^ stratified covariate Wald *p*-value by center.

## Data Availability

Data will be given through an anonymized Excel file in the Appendix A.

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
