# Peer review of "The Immunoscore in Localized Urothelial Carcinoma Treated with Neoadjuvant Chemotherapy: Clinical Significance for Pathologic Responses and Overall Survival"

_cancers, 2021, doi:10.3390/cancers13030494_

Round 1

Reviewer 1 Report

Well-written manuscript. Minor comments:

- In multivariable models for TTR and OS, neither histologic variants nor higher stages reached statistical significance. This is quite surprising. How the authors explain these findings?

- in Table 1 please add whether T and N stage refers to either clinical or pathological stage. If available, both should be reported both.

- please add how recurrence was defined

Author Response

Thank you very much for the time it took to read our work and your comments.

Well-written manuscript. Minor comments:

- In multivariable models for TTR and OS, neither histologic variants nor higher stages reached statistical significance. This is quite surprising. How the authors explain these findings?

For clinical stage, part of the answer to this question, answers next question too: Correct evaluation of initial stage is difficult in the context of trans-urethral resection of bladder. For T-stage, pathologist will only say: T2 minimum because trans-urethral resection of bladder does not allow to evaluate adjacent organs infiltration. Therefore, T3 and T4 are further assessed using CT-scan, which can underestimate infiltration. Regarding N-stage, this is usually only radiological staging with CT-scan which can once again underestimate pathological disease. We have added a comment on this in the manuscript. We believe this explains why the stage is not significant in multivariable analysis.

We added line 299:

Higher clinical stage was not statistically significant in multivariable analysis, which is probably due to the difficulty of correct staging using radiological evaluation. Although this represents routine standard-of-care daily practice, it likely underestimates some patients staging.  

For variant histologies, we believe there were too few patients in our retrospective series in order to obtain significance in multivariable analysis.

We added line 325:  

As these variants are rare, specific studies to these histological subtypes should be proposed in order to properly assess their impact.

- In Table 1 please add whether T and N stage refers to either clinical or pathological stage. If available, both should be reported both.

line 103, was added:

Clinical staging was done using CT-scans for T and N stage. For T-stage, pathologic report of TURB was used to indicate patient was at least T2 and further assessment was done using CT-scan.

- please add how recurrence was defined

line 105, added:

Recurrence was defined as either metastatic disease on radiological assessment if seen within the first year of follow-up or histologically confirmed if recurrence was seen after 12 months following surgery, or local relapse which was defined histologically confirmed muscle-invasive urothelial carcinoma.

Reviewer 2 Report

Elise F. Nassif et al., retrospectively investigated the role of IS in 117 patients treated by neoadjuvant chemotherapy for localized MIBC. IS included quantification of densities of CD3+ and CD8+ T-lymphocytes in the tumor center and margins of pre-treatment tumor samples and biopsies (ISb). High IS was positively associated with complete response and improved outcome including TTR and OS, giving it a potential predictive role before local treatment.

A this study is retrospective and some clinical data are missing, the authors stated, that these results warrant further evaluation in prospective trials before being implemented in clinical routine. Please comment on this statement and provide a  study outline, if possible.

Otherwise, I don´t have any complaints.

Author Response

Thank you for your kind review and comments.

We are actually going to evaluate immunoscore within a prospective trial which is taking place in France and randomizes neoadjuvant chemo-immunotherapy versus standard-of-care chemotherapy. This will allow us to prospectively evaluate the prognostic and predictive impact of IS to both modality treatments.

A specific dedicate trial would be optimal but does not seem realistic.